# The Effect of Exercise Intensity on Affective and Repetition Priming in Middle-Aged Adults

**DOI:** 10.3390/ijerph19169873

**Published:** 2022-08-10

**Authors:** Cristina Perez-Rojo, Jennifer A. Rieker, Soledad Ballesteros

**Affiliations:** Departamento de Psicología Básica II, Universidad Nacional de Educación a Distancia, C/Juan del Rosal, 10, 28040 Madrid, Spain

**Keywords:** affective priming, implicit memory, exercise intensity, middle-aged adults, physical exercise, repetition priming

## Abstract

Previous research has shown that physical exercise improves memory. In the present study, we investigated the possible effects of the intensity of physical exercise as a function of the affective valence of words on implicit memory. In the study, 79 young adult volunteers were randomly assigned to perform moderate- (50% VO_2_max) or high-intensity exercise (80% VO_2_max) on a stationary bike. Once the required exercise intensity was achieved, participants performed an affective and repetition priming task concurrently with the physical exercise. Both groups showed similar repetition priming. The moderate-intensity exercise group showed affective priming with positive words, while affective priming was not found in the high-intensity exercise group. Facilitation occurred in both groups when a negative target word was preceded by a positive prime word. Our results suggest that the positive effect of physical exercise on memory is modulated by the affective valence of the stimuli. It seems that moderate-intensity exercise is more beneficial for implicit memory than high-intensity exercise.

## 1. Introduction

Physical activity (PA) can be defined as any body movement produced by skeletal muscles that require energy expenditure. Both moderate- and vigorous-intensity physical activity improve health [1]. Regular PA helps to prevent and treat at least 25 chronic medical conditions such as heart disease, stroke, diabetes, and breast and colon cancer [2]. It also helps to prevent hypertension, overweight, and obesity and can improve mental health, quality of life, and well-being [3], as well as some cognitive functions [4]. Aerobic exercise has been related to an improved ability to cope with stress, anxiety, and depression [5,6]. Moderate-intensity exercise increases positive effects [7,8] and has also been associated with improvements in executive control [9] and cognitive performance [10]. A recent systematic review and meta-analysis showed that aerobic exercise interventions effectively improve episodic memory. Mixed-effects analyses demonstrated the positive effect on episodic memory among studies with a high percentage of females (65–100%) and participants with normal cognition. Subgroup analyses revealed the moderating effect of age (*p* = 0.027), particularly studies on participants with a mean age between 55 and 68 years, as opposed to 69–85 [11].

According to the transient hypo-frontality model [12,13], the higher the exercise intensity, the larger the demands for neuronal and metabolic resources. Thus, high-intensity PA might impair proper prefrontal cortex functioning and affect higher cognitive functions while exercising. On the other hand, according to cognitive–energetic models (e.g., Refs. [14,15]), high-intensity exercise acts as a stressor that elevates the arousal level above a cognition beneficial threshold, inducing poorer cognitive performance. Exercise increases epinephrine and norepinephrine levels in the bloodstream [16].

Other studies have investigated the effects of acute high-intensity interval training (HIIT) as a time-efficient strategy to improve physical health and executive functioning, e.g., Refs. [17,18]. HIIT is characterized by repeated sessions of brief, intermittent, and intensive exercise separated by short phases of recovery. A recent systematic review [19] indicated that the majority of HIIT analyses showed the positive effects of HIIT on executive function.

It seems that exercise intensity affects higher cognitive processes rather than low-level cognitive processes. For example, simple reaction time is not significantly affected by an increase in exercise intensity, whereas performance on choice reaction time tasks decreases with increasing exercise intensity [20]. Several studies have reported that exercise can affect reaction time. For example, Kashihara and Nakahara [21] found that vigorous exercise improves the choice reaction time, but only for the first 8 min after exercise. Exercise did not affect the percent of correct choices the subjects made. Collardeau et al. [22] found no post-exercise effect in runners but did find that exercise improved reaction time during the exercise, perhaps due to increased arousal. With high-level cognitive demands, the optimum performance occurs with exercise levels of moderate intensity and duration. Two meta-analyses investigated the effects of exercise intensity on several cognitive functions [23,24]. Moderate-intensity exercise, as opposed to low- or high-intensity exercise, has shown the facilitating effects on processing speed and, with lower effect sizes, on attention and memory recall. Furthermore, effects vary as a function of cognitive demands and moment of assessment, with larger effects associated with executive functions than in other cognitive domains. Longer time gaps between exercise and test administration also had larger effects. When test administration followed exercise immediately, only light and moderate-intensity exercise produced cognitive enhancement, whereas after a short delay, the benefits of light exercise disappeared, and an enhancement was found after the performance of moderate-intensity exercise [23].

Evidence on the relationship between exercise and implicit memory is scarce (for a review, see [25]). Implicit memory is usually measured by showing priming, which is understood as the involuntary response facilitation or inhibition produced by one stimulus (prime) upon another stimulus (target) presented afterward. Repeated stimuli are identified more quickly and/or more accurately than non-repeated stimuli in young adults and healthy older adults [26,27,28], as well as in cognitively impaired older adults [29,30,31,32]. Most investigations studied visual and auditory priming, but it also exists in touch [33,34] and taste [35].

Recently, in two experiments using a word completion task, Loprinzi et al. [36] investigated if exercise intensity can modulate implicit memory. Neither the effect produced by a 15-min self-selected “brisk walk” (moderate-intensity exercise) nor the one produced by a 15-min run at 80% of the maximum heart rate (high-intensity exercise) differed from those produced in their respective active control groups.

Stimuli can also be categorized as a function of emotional valence. Affective priming relates to response facilitation produced by the automatic activation of attitudes from memory [37]. This type of priming is defined as the difference in performance on congruent trials (prime and target words have the same emotional valence) and incongruent trials (prime and target words have different emotional valences) [38]. In affective priming, the response to the target word is modulated by the valence of the prime stimulus. Primes with positive or negative emotional valence might either inhibit or facilitate the subsequent response to the target stimuli [39,40,41,42,43]. For example, Yao and Wang [39] examined the affective priming effect in a lexical decision task and found that positive prime words facilitated the processing of subsequent related stimuli, whereas negative prime words did not. According to the affect-as-information approach [44], affective information can be represented as nodes in a semantic network, in which the emotional load of prime stimuli modulates both the activation and use of affective associations. Positive primes increased the accessibility of the associative network, whereas negative primes produced inhibition [45]. This might be explained by the neural networks that underlie emotional processing.

Besides the common brain areas involved in memory processing, emotional memory additionally implicates amygdala networks that store hedonic features, and whose stimulation modulates the arousal levels [46,47]. However, to enhance memory stimuli, a certain threshold of emotional arousal must be passed. Only memories with a high emotional charge seemed to affect cognitive performance [48].

Previous research demonstrated gender differences in priming affective words. While the left hemisphere seems to prime affective words quickly regardless of gender, gender differences are likely in the right hemisphere because affective word processing occurs slowly in males but rapidly in females. This gender difference may result from increased sensitivity to the emotional feature of affective words in females [49].

As far as we know, this is the first study conducted to investigate the effect of exercise intensity on implicit memory as a function of stimulus repetition and the emotional valence of the stimuli. So, the present study aims (1) to investigate the effect of moderate versus high-intensity exercise on implicit memory, assessed with repetition priming, and (2) to study the effect of exercise intensity on the implicit processing of emotional words assessed by an affective priming task.

## 2. Materials and Methods

### 2.1. Sample Size

We conducted an a priori power analysis using G*Power 3.1.9.6 (Franz Faul, Kiel, Germany) [50] to calculate the appropriate sample size. Using an α value of 0.05, a power value of 0.80, and a medium effect size (*f* = 0.25) for differences in implicit memory [51] as a function of exercise intensity, as well as two groups within the F-test family, a total sample size of 76 was required. Considering a possible dropout rate of about 5%, a total of 80 participants was sufficient to detect the main significant effects.

### 2.2. Participants

Eighty healthy middle-aged adults were recruited at sports and leisure facilities located in Madrid (Spain). Before the experimental session, the physical condition of the participants was assessed with the Rockport fitness walking test [52], which is an indirect measure of the maximum amount of oxygen that one can utilize during high-intensity exercise (VO_2_max). The test consists of walking as fast as possible for one mile (1609 m) on a flat ground using a heart rate monitor. At the completion of the distance, the heart rate and time of completion were recorded. VO_2_max was calculated using the formula [52]:$${\mathrm{VO}}_{2\max}\;\left(\mathrm{mL}/\mathrm{kg}/\min\right)=132.853+\left(6.315\cdot \mathrm{gender}\right)-\left(0.3877\cdot \mathrm{age}\right)-\;0.0769\cdot \mathrm{weight})-\left(3.2649\cdot \mathrm{time}\;\left[\min\right]\right)-\left(0.1565\cdot f\mathrm{inal}\;\mathrm{heart}\;\mathrm{rate}\right)$$
where gender = 1 for male and 0 for female.

Participants also completed the physical activity readiness questionnaire (PAR-Q) to determine the safety or possible risks of exercising based on their health history, current symptoms, and risk factors. Furthermore, the participants fulfilled the following inclusion criteria: they had a score of 17 or below on the Beck’s depression inventory [53,54], no physical contraindications to perform the required exercise, no current history of psychiatric or neurological disorders, no major surgery at least 6 months before the experiment, and normal or corrected to normal vision. One participant did not meet the inclusion criteria (a score above 16 on the depression inventory) and was excluded from further analysis. So, a final sample was composed of 79 healthy volunteers, who were randomly assigned to perform the priming task either at a moderate-intensity exercise at 50% of VO_2_max or a high-intensity exercise at 80% of VO_2_max. The moderate-intensity group was composed of 39 participants (16 males, $M_{age}$ = 35.58, SD = 7.88) and the high-intensity group of 40 participants (20 males, $M_{age}$ = 34.63, SD = 7.64). All participants individually performed their assigned exercise intensity while simultaneously performing the implicit memory task. The maximum oxygen consumption (VO_2_max), which is the highest rate at which oxygen can be transported and used during aerobic activities, was used to define the physical exercise load [55]. We considered moderate-intensity exercise at 50% of VO_2_max and high-intensity exercise at 80% of VO_2_max. [56]. The heart rate (HR) was calculated with the formula provided by Karvonen et al. [57]:$$\mathrm{Target}\;\mathrm{HR}\;\left(\%\right)=\left(\mathrm{maximum}\;\mathrm{HR}-\mathrm{resting}\;\mathrm{HR}\right)\times \mathrm{intensity}\;\left(\%\right)+\mathrm{resting}\;\mathrm{HR}$$

To calculate the maximum HR, we used the formula of Gellish et al. [58]:$$\mathrm{maximum}\;\mathrm{HR}=207-\left(0.7\times \mathrm{age}\right)$$

HR at rest was obtained by measuring the participant’s pulse after lying in the supine position for 10 min, remaining as relaxed as possible. Therefore, the resulting equations for the calculation of the exercise load for each group are as follows:

Moderate-intensity exercise:$$\mathrm{Exercise}\;\mathrm{at}\;50{\%\;\mathrm{of}\;\mathrm{VO}}_{2\max}=\left(\mathrm{maximum}\;\mathrm{HR}-\mathrm{resting}\;\mathrm{HR}\right)\times 0.5+\mathrm{resting}\;\mathrm{HR})$$

High-intensity exercise:$$\mathrm{Exercise}\;\mathrm{at}\;80{\%\;\mathrm{of}\;\mathrm{VO}}_{2\max}=\left(\mathrm{maximum}\;\mathrm{HR}-\mathrm{resting}\;\mathrm{HR}\right)\times 0.8+\mathrm{resting}\;\mathrm{HR})$$

Table 1 shows the demographic and physical baseline characteristics of the moderate- and high-intensity training groups. All participants gave their written informed consent. The study was conducted in accordance with the ethical guidelines of the 1975 Declaration of Helsinki.

### 2.3. Affective and Repetition Priming Task

For the affective priming task, we selected 320 emotion-related words (i.e., words that directly label an emotional state, such as “happy” or “fear”) and emotion-laden words (i.e., words that are associated with an emotion, e.g., “Thinking about holidays makes me feel happy”) from the Madrid Affective Database for Spanish [59]. Of these, 160 words had a positive valence, while the other 160 words had a negative valence. All words had a length of between 6 and 9 letters and similar arousal levels. For a description of the normative data, see Table 2.

Prime words were matched to target words as closely as possible for valence, arousal, length, and orthographic neighborhood. Initial letters and suffixes were matched when possible. The valence values were used to construct 160 word pair words, forming four experimental conditions: (1) 40 congruent positive trials, in which a positive target word was paired with a positive prime word; (2) 40 congruent negative trials, in which a negative target was paired with a negative prime; (3) 40 incongruent positive trials, in which a positive target was paired with a negative prime; and (4) 40 incongruent negative trials, in which a negative target was paired with a positive prime.

The words were presented in Courier-font 18-point lowercase letters in the centre of a black computer screen. Prime words were presented in white and target words were presented in yellow. Each trial started with the presentation of the fixation point in the center of the screen for 300 ms. Then, the prime (positive or negative) appeared for 200 ms, followed by a 300 ms blank screen. After a stimulus-onset asynchrony of 500 ms, the target (positive or negative) appeared and remained on the screen for 2000 ms. The trial ended with a 2000 blank screen, leaving a response window of 4000 ms. See Figure 1 for a schematic representation of the task paradigm. Responses to positive targets were mapped to the right index and negative targets were mapped to the left index, or vice versa, in a counterbalanced order across participants. Participants were asked to attend to the word presented in yellow (target) and to respond as quickly and accurately as possible by pressing the “+” key if the word contained a positive affective valence or the “−” key if the word contained a negative affective valence. The word pairs were presented randomly in two blocks of 160 trials each, with a pause of 60 seg in between. In the second block, the word order was inverted within each pair. Thus, participants responded in the second block to targets that had previously been presented as primes. Altogether, the experiment comprised 160 congruent trials (the prime and target words of which were both positive (80 trials) or negative (80 trials)) and 160 incongruent trials (half of which were positive prime and negative target words (80 trials) and the other half were negative prime and positive target words (80 trials)), yielding a total of 320 trials per run.

### 2.4. Procedure

Participants were assessed individually at the sports facility, where the temperature was kept constant at 21 degrees Celsius. For the pulse measurement, a Suunto Ambit 3 Sport heart rate monitor with a chest belt was used. While performing the experimental tasks, participants were pedaling on a Tomahawk S Series stationary bicycle. On the handlebar of the bicycle, a platform was installed with a VAIO Sony VPCEH laptop (15.5 inches) and a 2.20 GHz Intel Core processor, and data collection was managed with E-Prime 2.0 (Psychology Software Tools, Inc., Pittsburgh, PA, USA) experimental software. Participants maintained a constant distance of about 60 cm from the eyes to the screen. Two buttons of the computer keyboard were assigned for the responses: one for the right hand and one for the left hand. To avoid injuries, before starting the experimental tasks, participants performed a 5 to 10 min warm-up at a constant rhythm and resistance on the stationary bicycle, while reading the task instructions. The task began when the participants maintained their pulse uniform at the corresponding pulse rate (+/−10 beats), either at 50% of VO_2max_ or at 80% of VO_2_max. Once the experimental run started, they were asked to keep the pulse stable until the task had finished. In the case of the affective priming task, participants could pedal at a light pace for 1 min after the first block, which lasted approximately 15 min, and could return to their corresponding pulse at the start of the second block. The whole experimental session lasted approximately 60 min. 

### 2.5. Data Analysis

For all reaction time (RT) analyses, only correct trials were included. Trials with response latencies below 200 ms and above 1200 ms were excluded from the analysis. The RT-trimming procedure eliminated 8.8% and 7.1% of trials with targets with a positive valence, and 8.71% and 6.47% of trials with targets with a negative valence for the moderate- and high-intensity exercise groups, respectively. After data trimming, all distributions of response latencies showed acceptable levels of normality, homoscedasticity, and independence. There were no negative associations between error rates and RTs in any experimental condition, thus ruling out the possibility of a speed–accuracy trade-off. Error rates were analyzed using Mann–Whitney U and Wilcoxon signed-rank tests. A significance level of *p* < 0.05 was adopted for all statistical contrasts. All the data analyses were conducted with SPSS Statistic for Windows, version 28.0, Armonk, NY, USA, IBM Coorporation.

## 3. Results

### Affective and Repetition Priming

In the present study, affective priming is defined as the difference in performance between congruent trials (prime and target words have the same emotional valence) and incongruent trials (prime and target words have different emotional valences). The valence could be either positive or negative. The word pairs were presented in two runs; in the second run, the word order was inverted. Thus, participants responded in the second trial run to words that were presented as primes during the first run. Table 3 presents a summary of the response latencies and error rates, as per the experimental and exercise conditions for the affective priming task.

To analyze the effect of emotional valence and exercise intensity on the facilitation produced by congruency, we conducted a 2 (group: moderate- vs. high-intensity exercise) × 2 (valence: positive vs. negative) × 2 (congruency: congruent vs. incongruent) × 2 (block: first trial run vs. second trial run) mixed ANOVA approach on RTs as a dependent variable, with the group as a between-subjects factor and the valence, congruency, and block as within-subject factors. The main effect of valence was significant $\left(F_{\left(1,77\right)}=133.383,\;MSE=3962.332,\;p<0.001,\;\eta_{p}^{2}=0.632,\;1–\beta =1\right)$**,** indicating that the overall reaction times (RTs) were faster in response to positive words compared to negative words. This effect was not influenced by the exercise intensity, as shown by a non-significant valence × group interaction $\left(p=0.219\right)$. The main effect of the group was not statistically significant $\left(p=0.96\right).$ On the other hand, the main effect of congruency was significant $\left(F_{\left(1,77\right)}=10.355,\;MSE=837.328,\;p=0.002,\;\eta_{p}^{2}=0.119,\;1–\beta =888\right)$, suggesting that RTs were different for trials in which target and prime shared the same valence (congruent trial) than when prime and target had different valences (incongruent trial). A significant congruency × valence interaction $\left(F_{\left(1,77\right)}=35.16,\;MSE=595.76,\;p<0.001,\;\eta_{p}^{2}=0.314,\;1–\beta =1\right)$ suggested that the congruency effect was different for positive vs. negative words. Pairwise comparisons showed that RTs in response to negative words were faster when preceded by a positive prime $\left(\mathrm{mean}\;\mathrm{difference}=19\;\mathrm{ms},\;p<0.001\right)$, producing an inverted priming effect in incongruent trials with negative target words. Trials with positive target words produced a very small priming effect, without reaching statistical significance $\left(\mathrm{mean}\;\mathrm{difference}=-4\;\mathrm{ms},\;p=0.184\right)$, and the congruency × group interaction did not produce statistically significant results $\left(p=0.063\right)$. Finally, a significant three-way valence × congruency × group interaction $\left(F_{\left(1,77\right)}=5.778,\;MSE=595.76,\;p=0.019,\;\eta_{p}^{2}=0.07,\;1–\beta =0.66\right)$ suggests that this effect might differ in both exercise groups as a function of the emotional valence of the target word. Pairwise comparisons revealed that the moderate-intensity exercise group obtained a significant priming effect in response to positive words $\left(\mathrm{mean}\;\mathrm{difference}=-13\;\mathrm{ms},\;p=0.004\right)$, whereas this effect was absent in the high-intensity exercise group $\left(\mathrm{mean}\;\mathrm{difference}=5\;\mathrm{ms},\;p=0.258\right)$. Furthermore, both groups responded faster to negative words when they were preceded by a positive prime (mean difference =19 ms and 19 ms for moderate- and high-intensity exercise, respectively, p<0.001), producing the beforementioned inverted priming effect in incongruent trials with negative targets (see Figure 2).

Regarding repetition priming, measured as the difference in response latencies between the first presentation and the repetition of the word-pair list, overall performance improved in the second trial run compared to the first run $\left(F_{\left(1,77\right)}=17.244,\;MSE=3134.634,\;p<0.001,\;\eta_{p}^{2}=0.183,\;1–\beta =0.984\right)$, independently of the exercise intensity $\left(p=0.498\right)$. The repetition of the trial list had no significant influence on the differential response pattern to positive versus negative words $\left(p=0.673\right)$ in either exercise group $\left(p=0.77\right)$. The response facilitation produced by congruent trials slightly improved in the second trial run $\left(F_{\left(1,77\right)}=3.914,\;MSE=394.635,\;p<0.051,\;\eta_{p}^{2}=0.048,\;1–\beta =0.498\right)$ in both groups to a similar degree $\left(p=0.613\right)$, suggesting a relationship between the strength of the association between the prime and target and the magnitude of the priming effect. Neither the three-way block × valence × congruency interaction $\left(p=0.642\right)$ nor the block × valence × congruency × group interaction $\left(p=0.659\right)$ produced significant results. 

The analysis of the error rates showed a similar pattern for the moderate- and high-intensity exercise groups. Overall, participants committed less errors in congruent trials when the target and prime were positive [positive congruent: 5.49%, negative congruent: 8.13% $\left(z=-4.275,\;p<0.001,\;r^{2}=0.231\right)$], as well as in incongruent trials when the prime was positive [positive incongruent: 6.15%, negative incongruent: 8.78% $\left(z=-4.906,\;p<0.001,\;r^{2}=0.3\right)$]. Statistically significant group differences in error rates were not found in either positive $\left(z=-0.015,\;n.s.,\;r^{2}<0.001\right)$ or negative congruent trials $\left(z=-2.51,\;n.s.,\;r^{2}<0.001\right)$, nor in positive $\left(z=-3.7,\;n.s.,\;r^{2}<0.001\right)$, or in negative incongruent trials $\left(z=-0.281,\;n.s.,\;r^{2}<0.001\right).$

## 4. Discussion

The present study investigated the effects of the intensity level of PA (high or moderate) while healthy young adults performed an affective priming verbal task. Implicit affective memory was assessed with a verbal priming task which included positive and negative affective words. Repetition priming was assessed by comparing the performance of the two groups in the first and second blocks.

Our results showed the classical repetition priming effect previously reported using different types of stimuli such as words (e.g., [31,60]), as well as familiar (e.g., [61,62]) and unfamiliar (e.g., [63,64]) objects. We found that both groups of young adults benefitted from the repetition of the stimuli to a similar degree, suggesting that exercise intensity did not influence repetition priming.

As regards affective priming, only the moderate-exercise group showed a priming effect in response to positive words. Strikingly, both moderate- and high-intensity groups showed more response facilitation in incongruent trials than in congruent trials, suggesting that lexical emotional information might not be completely processed at the pre-attentive processing stage. This assumption is supported by the fact that the congruency effect slightly increased on the second trial run, signaling that a larger stimulus exposition can produce a larger affective priming effect. Furthermore, our results show that positive primes produced more response facilitation than negative primes, independently of the valence of the target word, and that RTs in response to negative words were faster when preceded by a positive prime. This inverted priming effect has been observed previously. For example, Rossell and Nobre [42] investigated affective semantic priming using a lexical decision task with neutral, happy, fearful, and sad affective categories. Their results indicated a priming effect for positive and neutral stimuli, whereas fearful stimuli produced a null effect, and negative information response inhibition. Sass et al. [40] also found that only positive primes produced priming in a lexical decision task. Moreover, in another study, Yao et al. [65] reported that only positive primes produced significant priming effects, whereas negative primes produced an opposite priming effect.

Bower [44] proposed an associative network with a different organization of emotional states represented as nodes in a semantic network, i.e., the asymmetric effects of positive and negative information, and suggested that positive relationships could be better elaborated and interconnected than negative ones [66,67]. Therefore, a positive prime might have a different effect on the processing of a target compared to a negative prime, as positive and neutral information share a similar semantic network, whereas negative information induces compensatory mechanisms that inhibit the propagation of associations between related concepts [44]. The processing of negative effects could reduce the optimization of information processing by requiring additional effort and time to process the target stimulus [42]. 

Only the moderate-intensity exercise group showed priming for positive words. It seems that high-intensity exercise interferes with the processing of positive information. One explanation could be that the processing of emotional valence goes in line with the physical stress level. High-intensity exercise causes discomfort, which could hinder the processing of dissonant (in this case positive) affective information. Our results are in line with those of Niven et al. [68], in that that affective valence during high-intensity continuous exercise and high-intensity interval exercise is consistently less positive than moderate-intensity continuous exercise, suggesting that they are less pleasurable. It seems that moderate-intensity exercise produces positive mood effects, indicating a greater priming in response to positive words. In contrast, high-intensity exercise produces a negative effect by inhibiting the evaluation of the target word [69,70].

It appears that priming is differentially influenced by exercise intensity and emotional valence, and that the processing of emotional information is modulated by the physical condition a person experiences when the information is processed. Overall, our results contribute to the knowledge of how physical exercise influences cognition and emotional processing. Nonetheless, more research is needed to fully understand the complex relationship between exercise intensity, metabolic resources, and the ability to process emotional information. Previous research suggests that complex neuromodulation, associated with extreme stress, might inhibit certain neural networks that are involved in the processing of higher cognitive functions.

## 5. Conclusions

In conclusion, moderate-intensity exercise seems to keep energetic resources at a cognition-beneficial threshold, whereas an increase in intensity goes beyond it. In the present study, moderate-intensity exercise was related to an enhancement in implicit affective memory, and this was not the case for high-intensity exercise. On the other hand, both groups showed repetition priming. The practice of high- and moderate-intensity exercise benefitted similarly from stimulus repetition, suggesting that exercise intensity did not affect implicit processing per se, but was limited to the implicit processing of emotional information.

## 6. Limitations and Future Directions

This study is not without limitations. Our study was a two-arm uncontrolled study, and the absence of a control group does not allow us to generalize the results. On the other hand, even though the sample was selected as homogeneously as possible, most of the participants were in good physical shape, and results should be replicated with less active adults. Another possible limitation refers to the higher intensities of physical exercise. From a certain intensity of exercise onwards, certain parts of the task, such as simply pressing a button, are hindered, or in some cases even by drying sweat, which can lead to longer RTs. The results of this study therefore show the effect of exercise on implicit memory in young adults. Physical exercise influences cognition in different ways at different ages of the life cycle. It would be interesting to extend the study to different populations, such as different age groups or clinical populations.

## Figures and Tables

**Figure 1 ijerph-19-09873-f001:**
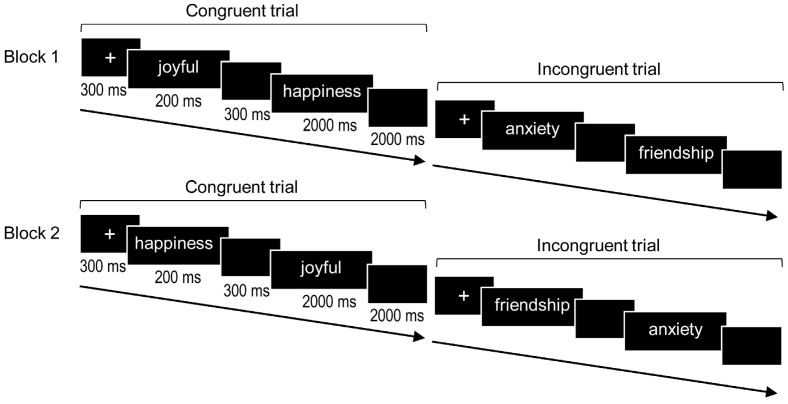
A schematic representation of the priming paradigm. In the first block, 80 positive and 80 negative word pairs, which could either be congruent (the valence of the prime word and the target word were the same) or incongruent (both valences differed), were presented. In the second block, primes and targets reversed. So, the prime words of the first block were now the target words, and targets of the first block were now primes.

**Figure 2 ijerph-19-09873-f002:**
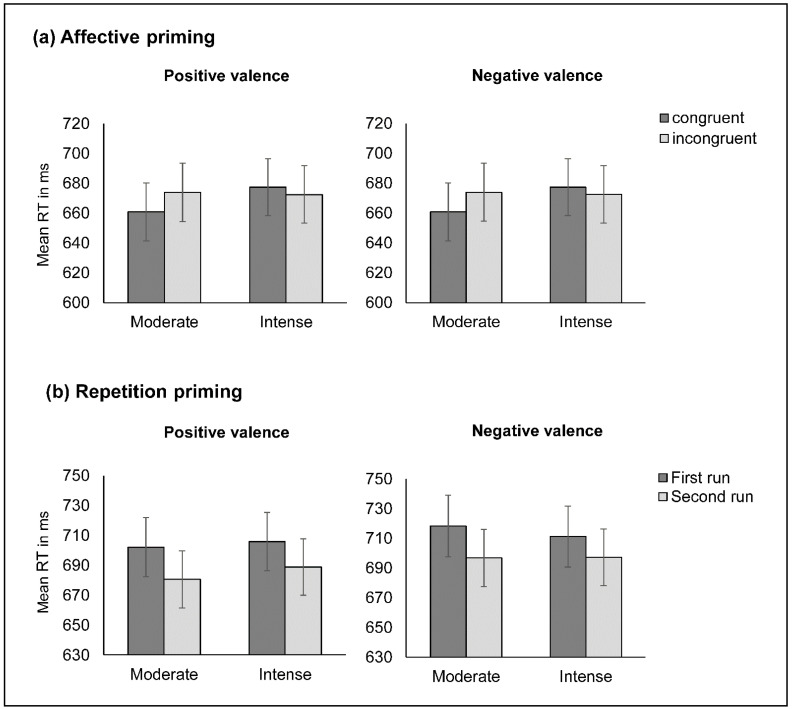
(**a**) Affective priming; (**b**) repetition priming as a function of exercise intensity and valence. Error bars: +/−1 SE.

**Table 1 ijerph-19-09873-t001:** Mean values of socio-demographic background variables for the moderate- and high-intensity exercise group. SDs are shown in parentheses.

	Moderate-Intensity (n = 39)	High-Intensity (n = 40)	*t* (df)	*p*
Men/women	16/23	20/20	*t* (77) = −0.794	0.430
Age	35.54 (7.78)	34.65 (7.74)	*t* (77) = 0.509	*0.612*
Education ^1^	1.38 (1.14)	1.58 (1.11)	*t* (77) = −0.754	0.453
Depression ^2^	4.77 (3.97)	4.03 (3.29)	*t* (77) = 0.908	0.367
VO_2_max ^3^	42.22 (6.53)	43.80 (7.90)	*t* (77) = −0.965	0.338

^1^ Level of educational attainment was defined as follows: 0 = junior high school, 1 = senior high school, 2 = short-cycle tertiary education, 3 = Bachelor’s or equivalent, ^2^ Beck’s depression inventory (BDI) [53]. ^3^ VO_2_max measured after the 1-mile walk using the Rockport fitness walking test [51].

**Table 2 ijerph-19-09873-t002:** Valence means and norming data ^1^ for positive and negative words. SDs are shown in parentheses.

	Positive Valence (n = 160)	Negative Valence (n = 160)
Valence	7.48 (0.59)	2.14 (0.61)
Arousal	6.01 (1.10)	6.32 (1.01)
Length (letters)	7.50 (1.10)	7.75 (1.01)
Length (syllables)	3.10 (0.69)	3.17 (0.57)
Grammar class (a/n/v) ^2^	38/55/67	27/59/74

^1^ Madrid Affective Database for Spanish (MADS) [59]. ^2^ a = adjective, n = noun, v = verb.

**Table 3 ijerph-19-09873-t003:** Descriptive data of the affective priming task for the moderate-intensity and high-intensity exercise group per target valence, prime–target affective congruency, and trial run. Means (standard deviations) for reaction times in ms and median values (interquartile ranges) for error rates.

Trial Run		Positive Valence		Negative Valence	
		Congruent	Incongruent	Congruent	Incongruent
		**Response Latencies**			
Moderate-intensity exercise	1	673.44 (129.70)	682.59 (130.10)	753.94 (139.62)	730.80 (131.57)
	2	648.25 (118.74)	665.35 (116.61)	728.21 (138.76)	712.85 (128.52)
High-intensity exercise	1	686.25 (122.25)	677.48 (130.97)	744.92 (131.38)	725.53 (122.46)
	2	668.64 (120.79)	667.59 (120.67)	726.95 (120.14)	709.13 (120.22)
		**Error rates**			
Moderate-intensity exercise	1	3.0 (0.0–7.0)	5.0 (3.0–7.0)	10.0 (5.0–12.0)	7.0 (5.0–15.0)
	2	3.0 (3.0–5.0)	5.0 (3.0–5.0)	5.0 (3.0–10.0)	7.0 (5.0–10.0)
High-intensity exercise	1	4.0 (0.0–10.0)	5.0 (0.7–10.0)	7.0 (3.5–10.0)	7.0 (3.0–11.5)
	2	3.0 (0.0–6.5)	4.0 (3.0–7.0)	5.0 (3.0–10.0)	7.0 (5.0–12.0)

## Data Availability

The raw data supporting the conclusions of this article will be made available by the authors, without undue reservation.

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
