# Peer review of "The Effect of Exercise Intensity on Affective and Repetition Priming in Middle-Aged Adults"

_ijerph, 2022, doi:10.3390/ijerph19169873_

Round 1

Reviewer 1 Report

The subject addressed in this article is worthy of investigation. The information presented was new. The conclusions were supported by the data. The methodology and statistical analysis were appropriate.

 1. Were the subjects assessed for body weight, height, BMI and waist circumference, Charlson coexistence index (CCI), and cardiopulmonary exercise test (CPET)?

CPET is an cardiopulmonary exercise test with respiratory gas analysis. Ventilation and gas exchange are measured breath after breath using an automatic metabolic measurement system. This allows a direct assessment of VO2 max.  The method used in the presented study is only an estimate.

2. The lack of a control group is a weak point of the study, which has already been pointed out by the authors. In future studies, e.g. in the elderly, it will be worth taking care of this element. Future studies, e.g. in the elderly, should incorporate this component.

 3. Most of the literature (48 items out of 65) is more than five years old. Please update with the more recent papers.

Author Response

Response to Reviewer #1 comments:

The subject addressed in this article is worthy of investigation. The information presented was new. The conclusions were supported by the data. The methodology and statistical analysis were appropriate.

 RESPONSE

 Thank you for your positive words.

 (1) Were the subjects assessed for body weight, height, BMI and waist circumference, Charlson coexistence index (CCI), and cardiopulmonary exercise test (CPET)?

CPET is an cardiopulmonary exercise test with respiratory gas analysis. Ventilation and gas exchange are measured breath after breath using an automatic metabolic measurement system. This allows a direct assessment of VO2 max.  The method used in the presented study is only an estimate

 RESPONSE

We appreciate your comment. We are aware that the method we used is an estimation of VO2 max.

(2) The lack of a control group is a weak point of the study, which has already been pointed out by the authors. In future studies, e.g. in the elderly, it will be worth taking care of this element. Future studies, e.g. in the elderly, should incorporate this component.

 RESPONSE

The Reviewer is right. This is a point to consider in future studies.

(3) Most of the literature (48 items out of 65) is more than five years old. Please update with the more recent papers.

 RESPONSE

We tried to cite relevant literature related to our study.

Reviewer 2 Report

The present study investigates the possible effects of the intensity of physical activity (PA as a function of the affective valence of words on implicit memory. The authors recruited 79 young adults who performed a moderate exercise (50% VO2max) or an intense exercise (80% 12 VO2max; concurrently with the physical exercise, the participants performed an affective and repetition priming task. Both groups (moderate PA and intense PA) showed similar repetition priming. The moderate PA group showed affective priming with positive words, while affective priming was not found in the intense exercise group. Facilitation occurred in both groups when a positive prime word preceded a negative target word. The results suggest that the positive effect of PA on memory is modulated by the affective valence of the stimuli, the moderate exercise being more beneficial for implicit memory than intense exercise.

The present study adds to our understanding of PA and its effects on cognition, extending previous research on this topic. The results are interesting and novel. The study is well conducted, the statistical treatment is appropriate, and the results are understandable. The discussion section is clear.

It is an original and well-written paper that provides a basis for further research regarding the effects of PA. However, I have some concerns regarding the strength of the present study:

1.      How does the research to Environmental Research and Public Health?

2.      What are the implications for clinical practice for these findings?

3.      Introduction (line 39-40). “Exercise increases 39 epinephrine and norepinephrine levels in the bloodstream” – the authors should add a reference

4.      Introduction (line 51): “Two meta-analyses investigated the effects of exercise intensity on several cognitive functions [18].” – The authors mention two research papers but provide one reference (which is not a systematic review)

5.      Line 66: “[…] healthy older adults [22-24], and in healthy and cognitively impaired older adults [25-28]” There is a repetition

6.      How were the participants recruited? Was it a random or consecutive sample?

7.      The authors acknowledge that the data is not generalizable; the cohort consisted of healthy young adults. However, the authors state that participants were "in good physical shape." How did they assess that?

8.      Previous research demonstrated gender differences in priming affective words. While the left hemisphere seems to prime affective words quickly regardless of gender, gender differences are likely in the right hemisphere because affective word processing occurs slowly in males but rapidly in females. This gender difference may result from increased sensitivity to the emotional feature of affective words in females.

-        Abbassi E, Blanchette I, Sirmon-Taylor B, Ansaldo AI, Ska B, Joanette Y. Lateralized Affective Word Priming and Gender Effect. Front Psychol. 2019;9:2591. doi: 10.3389/fpsyg.2018.02591.

9.      Recent systematic reviews showed that aerobic exercise interventions effectively improve episodic memory. Mixed-effects analyses demonstrated a positive effect on episodic memory among studies with a high percentage of females (65–100%) and participants with normal cognition. Subgroup analyses revealed a moderating effect of age (p = 0.027), with a significant effect for studies with a mean age between 55–68 but not 69–85.

-        Aghjayan SL, Bournias T, Kang C, Zhou X, Stillman CM, Donofry SD, Kamarck TW, Marsland AL, Voss MW, Fraundorf SH, Erickson KI. Aerobic exercise improves episodic memory in late adulthood: a systematic review and meta-analysis. Commun Med (Lond). 2022;2:15. doi: 10.1038/s43856-022-00079-7.

10.   These effects may not be generalizable in the long term.

11.   Several papers are reporting that exercise can affect reaction time. Welford (1980) found that physically fit subjects had faster reaction times, and both Levitt and Gutin (1971) and Sjoberg (1975) showed that subjects had the fastest reaction times when they were exercising sufficiently to produce a heart rate of 115 beats per minute. Kashihara and Nakahara (2005) found that vigorous exercise did improve choice reaction time, but only for the first 8 minutes after exercise. Exercise did not affect the percent of correct choices the subjects made. Collardeau et al. (2001) found no post-exercise effect in runners but did find that exercise improved reaction time during the exercise. They attributed this to increased arousal during the exercise.

12.   There may be a link between intelligence and reaction time (Deary et al. 2001). Among people of normal intelligence, there is a slight tendency for more intelligent people to have faster reaction times, but there is much variation between people of similar intelligence (Nettelbeck, 1980). The speed advantage of more intelligent people is most significant on tests requiring complex responses (Schweitzer, 2001).

13.   The authors do not mention the handedness of the subjects.

Author Response

Response to Reviewer #2 comments:

The present study investigates the possible effects of the intensity of physical activity (PA as a function of the affective valence of words on implicit memory. The authors recruited 79 young adults who performed a moderate exercise (50% VO2max) or an intense exercise (80% 12 VO2max; concurrently with the physical exercise, the participants performed an affective and repetition priming task. Both groups (moderate PA and intense PA) showed similar repetition priming. The moderate PA group showed affective priming with positive words, while affective priming was not found in the intense exercise group. Facilitation occurred in both groups when a positive prime word preceded a negative target word. The results suggest that the positive effect of PA on memory is modulated by the affective valence of the stimuli, the moderate exercise being more beneficial for implicit memory than intense exercise.

The present study adds to our understanding of PA and its effects on cognition, extending previous research on this topic. The results are interesting and novel. The study is well conducted, the statistical treatment is appropriate, and the results are understandable. The discussion section is clear.

It is an original and well-written paper that provides a basis for further research regarding the effects of PA. However, I have some concerns regarding the strength of the present study.

 RESPONSE

  We appreciate the positive comments of Reviewer 2.

(1) How does the research to Environmental Research and Public Health?

 RESPONSE

We think that the topic of our manuscript fit well with the research published in IJERPH.

(2) What are the implications for clinical practice for these findings?

 RESPONSE

The results of the present study have implications for clinical practice. The positive effect of physical exercise on memory function is modulated by the affective valence of the stimuli. More important, the study showed that moderate exercise is more beneficial for implicit memory than intense exercise.

(3) Introduction (line 39-40). “Exercise increases 39 epinephrine and norepinephrine levels in the bloodstream” – the authors should add a reference

 RESPONSE

We appreciate your valuable comment. We have included the following citation in our revised manuscript: Hassane Zouhal , Christophe Jacob, Paul Delamarche, Arlette Gratas-Delamarche. Catecholamines and the effects of exercise, training and gender. Sports Med 2008;38(5):401-23. https://doi.org/10.2165/00007256-200838050-00004.

(4) Introduction (line 51): “Two meta-analyses investigated the effects of exercise intensity on several cognitive functions [18].” – The authors mention two research papers but provide one reference (which is not a systematic review)

 RESPONSE

The references to the two meta-analyses [23, 24] were two lines below and we corrected this.

(5) Line 66: “[…] healthy older adults [22-24], and in healthy and cognitively impaired older adults [25-28]” There is a repetition

 RESPONSE

Thanks for your comment. We have corrected this.

(6) How were the participants recruited? Was it a random or consecutive sample?

 RESPONSE

As we have indicated in the text, the participants were recruited at sports and leisure facilities located in Madrid (Spain). It was a random sample.

(7) The authors acknowledge that the data is not generalizable; the cohort consisted of healthy young adults. However, the authors state that participants were "in good physical shape." How did they assess that?

 RESPONSE

The physical shape was objectively assessed with the estimated VO2 max measure and, subjectively, with the Physical Activity Readiness Questionnaire (PAR-Q), as indicated in lines 138 – 140.

(8)   Previous research demonstrated gender differences in priming affective words. While the left hemisphere seems to prime affective words quickly regardless of gender, gender differences are likely in the right hemisphere because affective word processing occurs slowly in males but rapidly in females. This gender difference may result from increased sensitivity to the emotional feature of affective words in females.

-        Abbassi E, Blanchette I, Sirmon-Taylor B, Ansaldo AI, Ska B, Joanette Y. Lateralized Affective Word Priming and Gender Effect. Front Psychol. 2019;9:2591. doi: 10.3389/fpsyg.2018.02591.

 RESPONSE

Thank you very much for your suggestion. We have included this very valuable reference in the manuscript.

(9) Recent systematic reviews showed that aerobic exercise interventions effectively improve episodic memory. Mixed-effects analyses demonstrated a positive effect on episodic memory among studies with a high percentage of females (65–100%) and participants with normal cognition. Subgroup analyses revealed a moderating effect of age (p = 0.027), with a significant effect for studies with a mean age between 55–68 but not 69–85.

-        Aghjayan SL, Bournias T, Kang C, Zhou X, Stillman CM, Donofry SD, Kamarck TW, Marsland AL, Voss MW, Fraundorf SH, Erickson KI. Aerobic exercise improves episodic memory in late adulthood: a systematic review and meta-analysis. Commun Med (Lond). 2022;2:15. doi: 10.1038/s43856-022-00079-7.

 RESPONSE

Many thanks to let us know about this recent and very relevant publication that we have included in the revised version of the manuscript.

(10) These effects may not be generalizable in the long term.

 RESPONSE

Reviewer 2 is right. Please, note that we referred to it in our revised manuscript.

(11) Several papers are reporting that exercise can affect reaction time. Welford (1980) found that physically fit subjects had faster reaction times, and both Levitt and Gutin (1971) and Sjoberg (1975) showed that subjects had the fastest reaction times when they were exercising sufficiently to produce a heart rate of 115 beats per minute. Kashihara and Nakahara (2005) found that vigorous exercise did improve choice reaction time, but only for the first 8 minutes after exercise. Exercise did not affect the percent of correct choices the subjects made. Collardeau et al. (2001) found no post-exercise effect in runners but did find that exercise improved reaction time during the exercise. They attributed this to increased arousal during the exercise.

 RESPONSE

Thanks for your very helpful suggestions. All these papers and others in fact maintained that exercise improves reaction time. We included the two more recent papers in the revised manuscript.

(12) There may be a link between intelligence and reaction time (Deary et al. 2001). Among people of normal intelligence, there is a slight tendency for more intelligent people to have faster reaction times, but there is much variation between people of similar intelligence (Nettelbeck, 1980). The speed advantage of more intelligent people is most significant on tests requiring complex responses (Schweitzer, 2001).

 RESPONSE

This is a very interesting comment, but we have not assessed the intelligence of our participants.

(13) The authors do not mention the handedness of the subjects.

 RESPONSE

Handedness was assessed and controlled for by counterbalancing the response mapping.

Round 2

Reviewer 2 Report

The authors correctly addressed my previous comments.